# Xylan Decomposition in Plant Cell Walls as an Inducer of Surfactin Synthesis by *Bacillus subtilis*

**DOI:** 10.3390/biom11020239

**Published:** 2021-02-08

**Authors:** Ida Szmigiel, Dorota Kwiatkowska, Marcin Łukaszewicz, Anna Krasowska

**Affiliations:** 1Faculty of Biotechnology, University of Wrocław, F. Joliot-Curie 14a, 50-383 Wrocław, Poland; ida.szmigiel@uwr.edu.pl (I.S.); marcin.lukaszewicz@uwr.edu.pl (M.Ł.); 2Faculty of Natural Sciences, Institute of Biology, Biotechnology and Environment Protection, University of Silesia in Katowice, Jagiellońska 28, 40-032 Katowice, Poland; dorota.kwiatkowska@us.edu.pl

**Keywords:** *Bacillus subtilis*, surfactin, hemicellulose, xylan

## Abstract

Hemicellulose is the second most abundant plant heterogenous biopolymer. Among products obtained from a wide range of agro-residues, biosurfactants, e.g., surfactin (SU), are gaining increasing interest. Our previous studies have shown that a *Bacillus subtilis* strain can successfully produce a significant amount of SU using a rapeseed cake. This work aimed to investigate plant hemicellulose components as substrates promoting SU’s efficient production by *B. subtilis* 87Y. Analyses of SU production, enzymatic activity and cell wall composition of hulled oat caryopses suggest that the main ingredients of plant hemicellulose, in particular xylan and its derivatives, may be responsible for an increased biosurfactant yield.

## 1. Introduction

Microbial hemi- and lignocellulosic hydrolysis of biomass has become increasingly important not only for reducing waste burden but also for producing a variety of high-value products [1,2,3,4]. Hemicellulose is the second most abundant plant heterogenous biopolymer. It is often composed of xylan [5,6], which has a backbone of β-(1→4)-linked xylose residues with side chains containing acetyl group, arabinose, or other sugars [7]. The precise structure of xylans varies between plant species and between different tissues [8]. Endo-β-1,4-xylanases and β-1,4-xylosidases degrade the xylan backbone while the xylan side-chains are hydrolyzed by α-l-arabinofuranosidase, α-glucuronidase, and acetylxylan esterase [9]. 

Among products obtained from a wide range of agro-residues, biosurfactants are gaining interest [10]. Biosurfactant surfactin (SU) is a well-known microbial-derived cyclic lipopeptide showing many useful properties. Among others, SU is known for its antimicrobial and antitumor activity [11,12]. Our previous studies showed that *Bacillus subtilis* successfully produced SU in solid-state fermentation (SSF) on rapeseed cake [13]. Moreover, during preliminary investigations we observed increased SU production on several types of cereals, especially milled oats. Production of SU by *B. subtilis* has been connected to the degradation of plant cell wall elements, especially xylan [14]. Hence, we decided to investigate the relationship between degradation of oat cell wall components and SU production. In this study, we used *B. subtilis* 87Y previously isolated from soilworm *Eisenia fetida* [15] We decided to use the SSF process, which provides many benefits. Compared to liquid state fermentation, SSF requires lower energy, produces less wastewater, is environment-friendly, and most importantly, it resolves the huge problem of solid waste or by-product disposal [16,17]. It also better reflects the natural environmental conditions of the *B. subtilis* interaction with plant material, i.e., the degradation of plant debris in the soil. In this study we performed SSF on mixtures of rapeseed meal (RSM) and milled oats (MO) along with RSM with xylan supplementation. To find out the connection between oat cell wall components and surfactin production we examined: xylanolytic activity, surfactin production, loss of hemicellulose content, expression of genes coding specific xylanolytic enzymes (endo-β-xylanase (*xynA*), β-xylosidase (*xynB*), α-arabinofuranosidase (*abfA*) and acetylxylan esterase (*cah*)) and carried out macro- and microscopic analysis of oat cell wall composition. 

## 2. Materials and Methods

### 2.1. Bacterial Strains, Culture Media, and Culture Conditions

*Bacillus subtilis* 87Y used in the study was previously isolated from the soilworm *Eisenia fetida* [15]. Bacterial inoculum with optical density OD600 = 0.1 was prepared by resuspending *B. subtilis* precultures in minimal MIM1 medium (sucrose 60 g/L, urea 2.3 g/L, MgSO_4_ 0.5 g/L, Na_2_HPO_4_ 8.4 g/L, NaH_2_PO_4_ 3.9 g/L, FeSO_4_ 1.2 mg/L, CuSO_4_ 1.6 mg/L, MnSO_4_ 5 mg/L, pH = 7.0). For the SSF we used a rapeseed meal (RSM), obtained from InventionBio (Bydgoszcz, Poland), and oats obtained from Agrolok (Golub-Dobrzyń, Poland). Hulled caryopses of oat (*Avena sativa*) were first milled using a grinder. Then, small fragments, mainly of starchy endosperm and aleurone layer, referred to as milled oats (MO), were separated with a coarse sieve from the larger fraction (LMO), which was enriched in hull (lemma and palea) fragments (Appendix A). RSM, MO, LMO, and hulled caryopses (HC) (Appendix A) were pasteurized for 30 min at 100 °C, cooled to room temperature and then mixed with inoculum in 1:1 ratio (*m*/*v*) to maintain 50% humidity. SSF was performed in 100 mL Erlenmeyer flasks for 24 h at 37 °C. After 24 h the biomass was immediately frozen at −80 °C and air-dried. Dry biomasses were used for analyses of fiber content, enzymatic activity, SU production, and cell wall composition. The treatment in which the inoculation step was omitted was used as a control for the cell wall analysis.

### 2.2. Determination of Neutral Detergent Fiber (NDF) and Acid Detergent Fiber (ADF) Content in the Fermented Biomass

Analysis was performed in Laboratory of Department of Animal Nutrition and Feed Management of Wroclaw University of Environmental and Life Sciences. Determination of Acid Detergent Fiber (ADF) and Neutral Detergent Fiber (NDF) was performed using van Soest and Wine method [18] with ANKOM 2000 Fiber analysis device. Briefly, 0.5 g of air-drier sample was ground and passed through 40 mesh. Then, 100 mL of neutral detergent fiber solvent or acid detergent fiber solvent was added and boiled for 10 min. Samples were rinsed with hot water, filtered, and washed again. Then, crucibles were washed twice in acetone, dried overnight, and weighed. 

### 2.3. Xylanase Activity Assay

For xylanase activity determination, 3,5-dinitrosalicylic acid (DNS) method [19] and Bradford assays were performed. Liberation of reducing sugars in the culture filtrate was performed according to [20] with modifications. Reaction mixture containing 0.1 mL of appropriately diluted culture filtrate with 0.1 mL of 1% birchwood xylan (Serva) solution in acetate buffer (0.05 M, pH = 5.0) was incubated for 30 min at 50 °C. The reaction was stopped by the addition of 0.65 mL DNS and heated for 10 min at 100 °C. Then, the mixture was cooled down to room temperature, and the liberated reducing sugars were measured spectrophotometrically at 530 nm and expressed as xylose equivalent. Xylose was taken as standard. For protein determination, Bradford assay was performed according to Jones (2012) with modifications [7]. 0.5 mL of Bradford reagent was mixed with 0.5 mL of appropriately diluted culture filtrates and incubated for 10 min at room temperature. Protein content was measured spectrophotometrically at 595 nm. Protein concentration was calculated using bovine serum albumin (BSA, SERVA) as a standard.

Specific enzyme activity was defined as the amount of enzyme required to produce 1 µmole of xylose per minute under assay conditions per soluble protein in the culture filtrate (U/mg protein). 

### 2.4. Surfactin Production

One gram of lyophilized fermented biomass was added to 20 mL of methanol LC-MS (VWR). Samples were shaken at room temperature for 30 min with agitation of 180 rpm. Then, 1 mL of supernatant was centrifuged at 13,500 rpm for 10 min, and filtered through 0.22 µm. Supernatants were then used for UHPLC - MS system (UHPLC with QDa detector, Aquity ARC Waters) analysis, equipped with a CORTEX C18 column (4.6 × 50 mm; 2.7 µm). During analysis, the column was kept at 40 °C, samples at 15 °C. Mobile phases of water with 0.1% (*v*/*v*) TFA (A) and ACN with 0.1% (*v*/*v*) TFA were used. MS analysis was conducted in positive mode ESI. The source temperature was set to 600 °C. Nitrogen was used as desolvation gas. The cone voltage was set to 25 V, and the capillary voltage was set to 0.8 kV. Samples were analyzed in the range of 990–1080 *m*/*z*. 

### 2.5. Whole Genome Sequencing

The genome of B. subtilis 87Y was isolated using GeneMatrix Bacterial and Yeast Genomic DNA Purification Kit (EurX) and kept at −20 °C. Whole genome sequencing was performed by Eurofins Genomics (Germany) which comprised DNA fragmentation, adapter ligation, size selection, and amplification. The standard genomic library was prepared including unique dual indexing. Sequencing was performed using Illumina technology with pair end run type. Read length was 2 × 150 bp. *B. subtilis* 168 genome (NCBI: NC_000964.3) was used as a model species.

### 2.6. Sequence Alignments

Genes coding specific xylanolytic enzymes: endo-β-xylanase (*xynA*), β-xylosidase (*xynB*), α-arabinofuranosidase (*abfA*), and acetylxylan esterase (*cah*) from the *B. subtilis* 168 strain, were obtained from the National Centre of Biotechnology Information (NCBI) database. Accession IDs are: *xynA* (NC_000964.3), *xynB* (NC_000964.3), *abfA* (NC_000964.3) and *cah* (NC_000964.3). Sequence alignments were performed by EMBOSS Needle program. 

### 2.7. Quantitative PCR (qRT-PCR)

After 24 h of SSF, whole biomass was thoroughly washed with 0.9% NaCl for 30 min. Supernatant with cell suspension was then used for RNA isolation with RNA Total mini kit (A&A Biotechnology) according to manufacturer’s protocol. The concentration of RNA samples was determined spectrophotometrically using NanoDrop 2000 (Thermo Fisher Scientific). Samples were treated with DNAse I (Fermentas) to remove genomic DNA contamination. The cDNA was synthetized using 1 µg RNA with a High-Capacity cDNA Reverse Transcription Kit (Applied Biosystems). 

Serial dilutions of cDNA were used to generate calibration curves and calculate PCR amplification efficiencies (Table 1). Specific primers for *gyrB, xylA, xylB, abfA*, and *cah* genes were used (Table 1). Real-Time PCR reaction was performed with iTaq Universal SYBR Green Supermix (Bio-Rad) and Step-One Plus Real-Time PCR System (Applied Biosystems). The thermal cycling conditions consisted of the initial step at 95 °C for 10 min, followed by 40 cycles at 95 °C for 20 s, 57 °C for 20 s, and 72 °C for 30 s. The *gyrB* gene was selected using Reffinder tool as the most stable housekeeping gene. Relative expression of target genes was calculated using Pffafl method, taking into account PCR amplification efficiencies.

### 2.8. Macroscopic and Microscopic Analyses of Oats

Size fractions of MO were determined for 5–10 mg samples of pasteurized, fermented, and lyophilized (PFL), and pasteurized, non-fermented, and lyophilized (PNFL) biomass using calibrated sieves (Multiserw-Morek, Poland). Images of various fractions were obtained using stereoscopic microscope (Nikon SMZ 288) equipped with digital camera (DS-Fi1). 

Microscopic analyses were performed on MO (PFL, PNFL) and HC (PFL, PNFL, I). Three types of samples were observed in the confocal microscopy: transverse sections through caryopsis and transverse sections through lemma, made with a razor blade; and whole mount fragments of MO. The samples were first placed in a blocking buffer containing 3% bovine serum albumin in PBS for 1 h. Next, the sections were incubated overnight with the primary antibody LM11 (PlantProbes) at 4 °C [21], rinsed in PBS (three times for 10 min), and incubated for 1 h with the secondary antibody conjugated with fluorochrome (Alexa Fluor 488), in darkness at room temperature. Following the incubation, the samples were rinsed in PBS (three times for 10 min), counterstained with 0.01% Calcofluor White (Fluorescent Brightener 28, Sigma-Aldrich) in PBS solution, and again rinsed with PBS (five times for 5 min) [22]. Such prepared samples were mounted on glass slides in Floromount (Sigma-Aldrich). The fluorescence of the Calcofluor White (excitation 365 nm, emission 435 nm) and of the secondary antibody (excitation 490 nm, emission 525 nm) were detected using Olympus FV-1000 confocal system equipped with Olympus IX 81 inverted microscope, a 405 nm diode laser, and a multi-line (458 nm, 488 nm, 515 nm) argon ion laser (Showa Optronics Co., Ltd, Tokyo, Japan). 

Stacks of confocal images were analyzed using ImageJ Fiji (National Institutes of Health, Bethesda, MD, USA; https://imagej.nih.gov/ij/ (accessed on 10 December 2020)). Plots were generated in Matlab (Mathworks, Nattick, MA, USA).

### 2.9. Statistical Analysis

At least three independent replicates were performed for each experiment. Statistical significance was determined by Student’s *t*-test using GraphPad Prism or Matlab software.

## 3. Results and Discussion

### 3.1. Production of Surfactin and Activity of Xylanase Depend on Hemicellulose Content in Cell Walls

Our preliminary results on SSF of various cereals showed the highest SU yield on milled oats (MO) (data not shown). Hence, we decided to study MO as a material increasing biosurfactant production. We performed SSF on RSM as control and compared the obtained results with SSF on RSM, a part of which was replaced by an increasing amount of MO (50% and 100% MO), by measuring the SU content in the extracts from all the samples.

According to our previous study, *B. subtilis* strains can produce various analogues of SU during growth on a rapeseed cake [13]. *B. subtilis* 87Y strain used in this study can successfully produce approximately 1.8 g of SU per kg of dry RSM (Figure 1A). The addition of MO increases the obtained SU yield to about 3.3 g of SU per kg of dry MO (Figure 1A). Various agro-industrial residues, such as cassava wastewater, sludge palm oil, distillery, and whey wastes or lignocellulosic wastes, have already been used for biosurfactant production [23], also specifically for the SU production [24,25,26,27]. According to our knowledge, MO has not yet been tested for SU production. The highest yield of SU from agro-based substrates was obtained by Gurjar and Sengupta (2015), who obtained 4.17 g/kg from rice mill polishing residue [28]. However, this amount was obtained after multi-stage foam fractionation, while our method assumes only simple extraction with MeOH. Interestingly, although *B. subtilis*, in addition to SU, can produce other lipopeptides, such as iturin and fengicin [29], in our experimental conditions, we did not observe other lipopeptides than SU.

*B. subtilis* 87Y produces about three times more xylanases on MO than during RSM fermentation (Figure 1A). However, we did not observe significant changes in the number of xylanases in RSM and MO mixtures (Figure 1B).

Because we observed increased SU production and xylanase activity in response to MO content in SSF growth medium, we decided to analyse the hemicellulose content in unfermented and fermented biomass (Table 2). NDF and ADF analyses revealed that unfermented MO contains much more hemicellulose than RSM (Table 2). 

Hemicellulose loss during the fermentation also increased with increasing MO content in solid medium used for SSF (Table 2), simultaneously with increasing SU production in these samples (Figure 1). Thus, we hypothesized that hemicellulose content in general or the content of a hemicellulose ingredient have an impact on biosurfactant production. 

### 3.2. Xylan Addition Affects Surfactin Production or Xylanase Activity during SSF

Plant hemicellulose contains xylan in relatively high amounts [30]. Previous studies have shown that cell wall polysaccharides of tomato and *Arabidopsis* roots enhance SU synthesis by *Bacillus amyloliquefaciens* strains [14]. An addition of only 0.1% cell wall polysaccharides, such as xylan or cellulose, resulted in increased SU production [14].

Because of the observed increase in xylanase activity and loss of hemicellulose in the tested samples, next we checked if preculture enhancement with xylan will affect SU production or xylanase activity. Since analysis of fibre composition revealed that MO contains noticeably higher amount of a hemicellulosic fraction, we changed carbon source from sucrose to xylan in *B. subtilis* inoculum for SSF. This study showed that although a growing level of xylan supplementation resulted in increased SU production (RSM_X1, RSM_X2, RSM_X4), the absolute quantity of obtained biosurfactant was lower than in the basic conditions (RSM_100, Figure 2A). In contrast to that, the addition of pure xylan did not drastically change the general xylanase activity (Figure 2B). The reason may be that usually xylanases are induced by pure xylan or xylan-rich residues [31]. However, it was also shown that among *B. subtilis* xylanases, only β-xylosidase was induced by xylan [32]. Furthermore, Schmiedel and Hillen (1995) have shown that *B. subtilis* 168 cannot grow using xylose as a sole carbon source [33]. However, our previous study showed that *B. subtilis* 87Y strain utilizes various sugars, including xylose [15].

Enriching RSM with sucrose and xylan (samples SX, Figure 3A) resulted in increased SU synthesis accompanied by growing xylanases activity. Only in samples enriched with the highest level of xylan (RSM_X4, Figure 3B), we did observe SU production level comparable to primary conditions with sucrose as the main carbon source (RSM_100, Figure 3A). Sucrose addition to fermentation partly restored the obtained SU yield, probably due to its easier decomposition and assimilation. Xylanolytic activity increment may result from the fact that DNS method is based on determining the amount of reducing sugars, which will be higher after fermentation with sucrose and xylan simultaneously than with xylan alone. 

### 3.3. Milled Oats and Pure Xylan Increase Expression of Specific Xylanolytic Enzymes

As the activity of xylanases rises during SSF on MO and on RSM with xylan supplementation, we decided to analyse the expression of gene encoding specific xylanases of *B. subtilis* 87Y. For this purpose, we sequenced the genome of *B. subtilis* 87Y and performed quantitative PCR. Efficient hydrolysis of xylan is performed by enzymes with diverse specificity. Endo-1,4-β xylanase (E.C. 3.2.1.8.) randomly cleaves the backbone of xylan; β-xylosidase (E.C. 3.2.1.37) cleaves monomers of xylose; while α—arabinofuranosidase (E.C. 3.2.1.55), acetylxylan esterase (E.C. 3.1.1.72) and α—glucoronidase (E.C. 3.2.1.139) are responsible for removal of the side groups attached to the xylan backbone [34]. In the *B. subtilis* 87Y genome we found the first four genes encoding xylanolytic enzymes and investigated their expression during SSF (Figure 4).

While in reference conditions (RSM_100) expression of genes encoding specific xylanolytic enzymes changed with minor differences, mixing RSM with MO significantly diversified pattern of activated genes (Figure 4A). Depending on quantity of added MO, we observed a significant rise of *xylB* and *cah* genes. Acetyl xylan esterase (*cah)* removes O–acetyl groups from β–D–xylopiranosyl residues of acetyl xylan [35]. Synergistic action of depolymerizing and side-group-cleaving enzymes was earlier demonstrated [35]. Xylanases have limited access to xylan backbone in the absence of esterases due to high degree of acetylation. Thus, acetyl esterase acts prior to the action of xylanases [35]. In turn, expression of *abfA* gene coding α–arabinofuranosidase that cuts off arabinose from arabinoxylans, decreases with higher amount of MO added to base (Figure 4A). Activation level of *abfA* is surprising, since *Avena sativa* is known for quite high content of arabinoxylans [36]. In contrast to that, addition of pure xylan to RSM causes increase of *xylB* gene coding β–xylosidase (Figure 4B). As mentioned earlier, it was shown that addition of xylan led to activation of β–xylosidases rather than endo–β–xylanases [32], which we also observed during the experiment (Figure 4B). 

### 3.4. Fermentation of Oats Affects the Cell Wall Content and Integrity

To elucidate the effect of MO addition to RSM on the SU production, we examined how cell walls of oat hulled caryopsis (HC) are affected by fermentation. First, comparing the contribution of three size fractions in MO before and after fermentation, we showed that the contribution of smaller fractions is increased (doubled or tripled) in fermented material, i.e., the size of tissue fragments decreases in the course of fermentation (Figure 5).

Next, we aimed at identifying main cell wall polysaccharides that are affected by fermentation. Knowing the activity of bacterial enzymes and specific composition of grass cell walls [30,37], we focused on detection of xylans (heteroxylans). Direct identification of polysaccharide epitopes using antibodies was disabled in MO by non-specific adherence of the secondary antibody to starch grains (Appendix A), the large quantities of which were present on the surface of milled tissue fragments (Appendix A). Thus we examined oat HC that were not milled, using antibody LM11, specific for unsubstituted and low substituted β-1,4-xylan [21], and Calcofluor White. Calcofluor binds to various β-glucan structures [22], including cellulose and mixed linkage glucans, i.e., (1,3;1,4)-β-D glucans, the hemicellulose specific for cell walls of grass endosperm [37]. 

In caryopsis, the LM11 and Calcofluor signal domains are complementary rather than overlaid (Figure 6). LM11 signal is confined to outer cell walls of the aleurone layer of endosperm, where outer periclinal walls (walls parallel to the organ surface) and portions of anticlinal walls (perpendicular to the organ surface) close to the endosperm surface have distinct LM11 signal localized at the older (outer) wall layers. Intact hulled caryopses (I-HC) differ from pasteurized, non-fermented, lyophilized (PNFL) and pasteurized, fermented, lyophilized (PFL) hulled caryopses (HC), in that the LM11 signal in I-HC reaches deeper parts of anticlinal walls of aleurone cells. Difference between PNFL- and PFL-HC is mainly in the shape of LM11 domains visible in cross sections. In PNFL-HC it resembles I-HC, i.e., the LM11 domain, which complements Calcofluor-labelled wall layers, is nearly continuous and has a triangle-like shape filling creases above the anticlinal walls, while in PFL-HC sections the LM11 domain is less regular and fragmented. We conclude that during the SSF some of LM11 epitope is likely removed, and integrity of outer cell walls of caryopses is affected. 

Because in this experiment the entire HC, i.e., caryopses covered with lemma and palea, were fermented, strong effects of fermentation were not expected in deeper layers of endosperm. Indeed, signal intensity of Calcofluor, the fluorescence of which was shown to increase after heteroxylan removal from cell walls [38,39,40], was similar in endosperm of PNFL- and PFL-HC (data not shown) although it was significantly lower in I-HC (most likely due to thermic and humidity treatments during pasteurization). Thus, next we examined cell walls of lemmas. LM11 signal in lemmas was present in all of the samples and came mainly from primary cell walls (Appendix A). Differences were in the Calcofluor signal intensity of thick secondary cell walls (Figure 6), which was significantly increased in PFL-HC lemmas in comparison with lemmas of PNFL- and I-HC (Figure 7). Apparently some of hemicellulose, most likely xylan which is abundant in secondary cells walls of grasses [30,41] is removed from lemma walls during fermentation. This is in agreement with experiments performed with hemicellulose loss within fermented RSM/MO mixtures (Table 2) and also the increased xylanolytic activity in SSF with MO as a solid base (Figure 1).

### 3.5. Components of Oat Hulls Induce Surfactin Production

To investigate in more detail the above results on the hemicellulose loss and xylanase activity, we examined the SU production on variously prepared oat kernels, comparing results of SSF on MO with SSF on LMO and whole HC (Figure 8). 

SU yield significantly decreased when fermentation was performed on HC. Although the same SSF process was performed (i.e., solid medium was pasteurized, fermented, and lyophilized-air-dried) the whole HC were not mechanically processed prior to fermentation, unlike MO. We expect that the grinding process makes the components of cell walls more accessible for bacteria. The interesting phenomenon is that SU yield on LMO (~4 g/kg of dry biomass) is even higher than on MO (~3.5 g/kg of dry biomass). Similar results were obtained by Slivinski et al. (2012) using *B. pumilus* strain during fermentation of okara with the addition of sugarcane bagasse [42]. Cultivation of *B. pumilus* in column bioreactor resulted in 3.3 g SU per kg of dried biomass. In turn, Zhu et al. (2013) used fermentation of rapeseed meal, corn meal, soybean flour, wheat bran, bean cake, and rice straw or their mixtures for the SU production [43]. The highest yield of SU (6.25 g/kg dry biomass) was obtained on rice straw with soybean flour [43]. Milled oats are a mixture of small fragments of oat hulls and starchy endosperm (the oat flour), while LMO is enriched in hull fragments. Thus, we suspect that starch grains may inhibit the production of SU by *B. subtilis*. At the same time microscopic analyses of oat cell walls revealed that hemicellulose fraction decreased during the fermentation (Figure 6). Oat hulls (lemma and palea) are hemicellulose–rich portion of oat kernels [44]. Along with the highest SU production on oat hulls, these results suggest that hemicellulose in general, or its components, like xylan and its derivatives, are likely the carbon source used by *B. subtilis* for biosurfactant production.

## 4. Conclusions

The presence of hemicellulose, especially the heteroxylans, the relatively high content of which is characteristic for grass cell walls, affects the SU production by *B. subtilis*. Along with higher yield of SU obtained with fermentation of milled oats we observed increased xylanase activity. Additionally, supplementation of SSF with both sucrose and xylan resulted in similar effect. We suspect that diverse fraction of oat heteroxylans are responsible for higher production of SU because we observed increased gene expressions of different specific xylanolytic enzymes. Additionally, SSF performed on hulled caryopses of oats led to a decrease in the hemicellulose content in the hull and lowered integrity of the heteroxylan-rich layers of aleurone cell walls. Since agro-residuals are mainly based on plant tissues abundant in hemicellulose, our results suggest that generally available resources can be used to obtain high value-added products through profitable fermentation process.

## Figures and Tables

**Figure 1 biomolecules-11-00239-f001:**
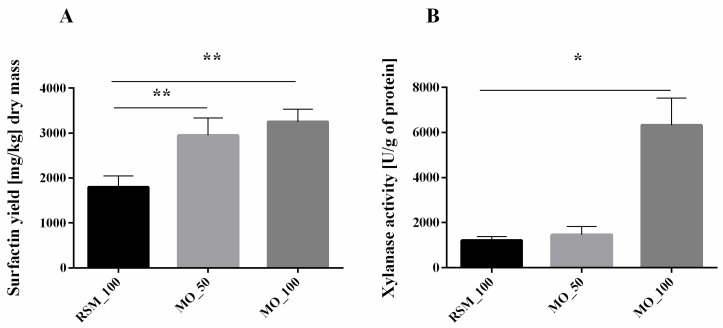
Surfactin yield (**A**) and general xylanase activity (**B**) obtained during 24 h solid-state fermentation (SSF) using *B. subtilis* 87Y. RSM—rapeseed meal, MO—milled oats followed by a number specifies the level of RSM replacement by MO. Statistical significance in all cases is presented as follows: * *p* < 0.05; ** *p* < 0.01.

**Figure 2 biomolecules-11-00239-f002:**
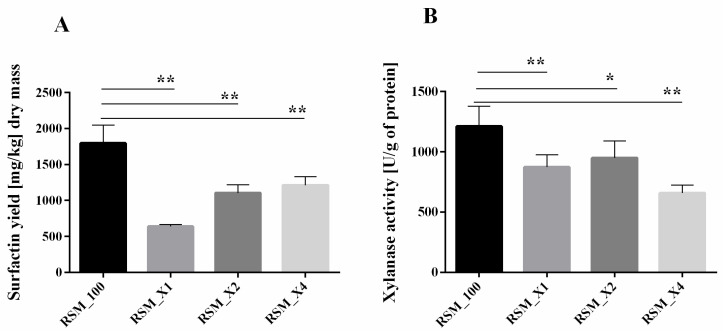
Surfactin yield (**A**) and general xylanase activity (**B**) obtained with *B. subtilis* 87Y during 24 h SSF on RSM supplemented with xylan as carbon source. X1, 2, 4—xylan addition in 10, 20 or 40 g/L. Statistical significance in all cases is presented as follows: * *p* < 0.05; ** *p* < 0.01.

**Figure 3 biomolecules-11-00239-f003:**
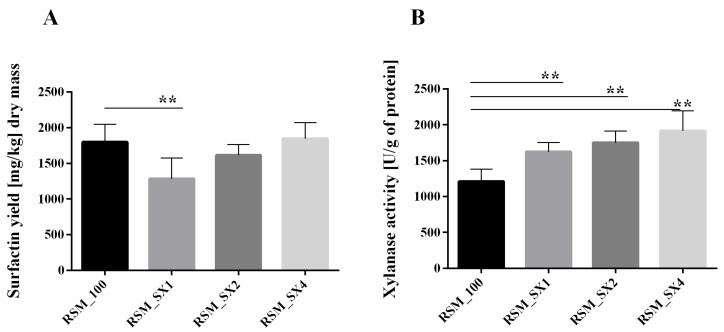
Surfactin yield (**A**) and general xylanase activity (**B**) obtained by *B. subtilis* 87Y during 24 h SSF on RSM with sucrose and xylan as carbon sources. SX1, 2, 4—sucrose addition in 60 g/L, xylan addition in 10, 20, or 40 g/L. Statistical significance in all cases is presented as follows: ** *p* < 0.01.

**Figure 4 biomolecules-11-00239-f004:**
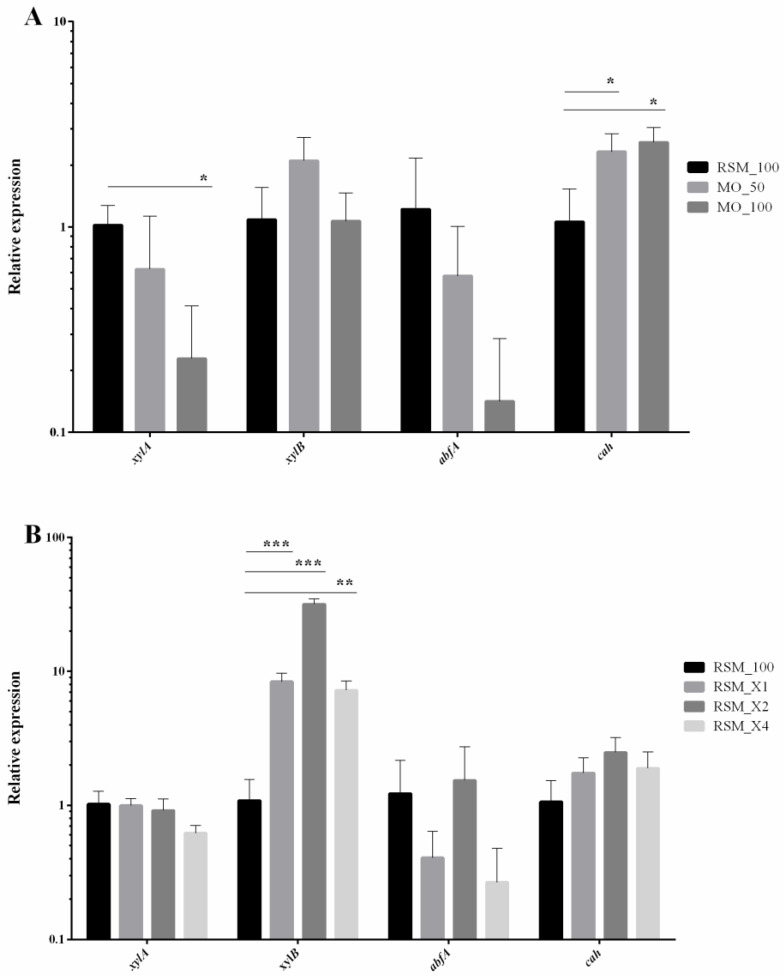
Relative expression of specific xylanases found in *B. subtilis* 87Y genome. (**A**) SSF samples on RSM/MO mixture; (**B**) SSF samples on RSM with xylan substitution (instead of sucrose) in liquid medium as carbon source of 10, 20 or 40 g/L. *xylA*—endo-β-xylanase; *xylB*—β–xylosidase; *abfA*—α–arabinofuranosidase; *cah*—acetylxylan esterase. Results are shown in log_10_ values. Statistical significance in all cases is presented as follows: * *p* < 0.05; ** *p* < 0.01; *** *p* < 0.001.

**Figure 5 biomolecules-11-00239-f005:**
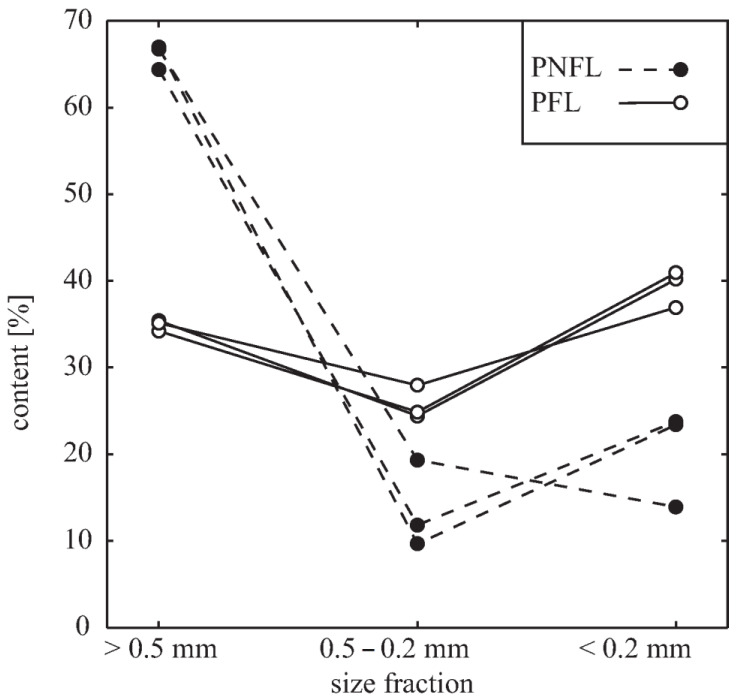
Contribution of various size fractions [*w*/*w*] in PFL- and PNFL-MO samples.

**Figure 6 biomolecules-11-00239-f006:**
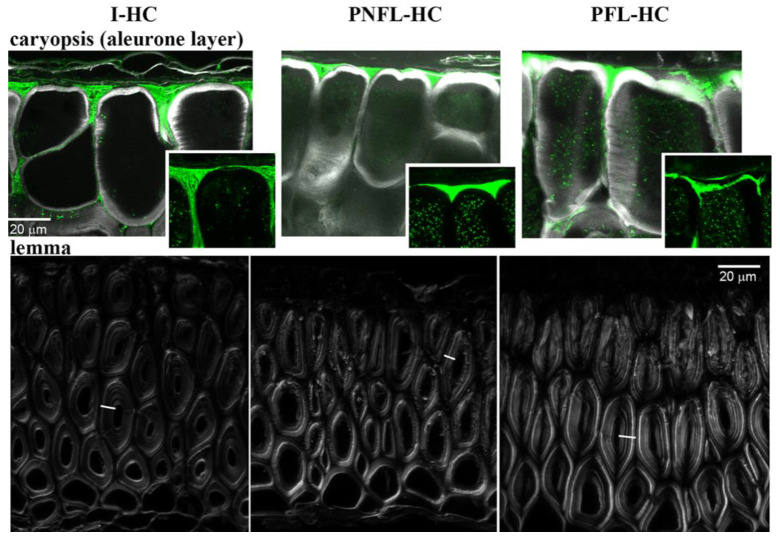
Confocal images of cross-sections of caryopsis aleurone layer (upper row) and lemma (lower row). Z-projections of maximal signal from confocal stacks were obtained using Fiji. For aleurone layer fragments, the overlaid LM11 (green) and Calcofluor White (grey) signals or only LM11 signal (insets) are shown. The dot-shaped LM11 signal from cell lumen likely results from nonspecific adherence of secondary antibody to cytoplasm organelles. In lemma cross-sections, thick-walled fibres were stained in Calcofluor White (grey) only. Line segments across the cell walls point to exemplary transects, shown in Figure 7, along which the grey signal intensity was measured. All images are shown at the same magnification.

**Figure 7 biomolecules-11-00239-f007:**
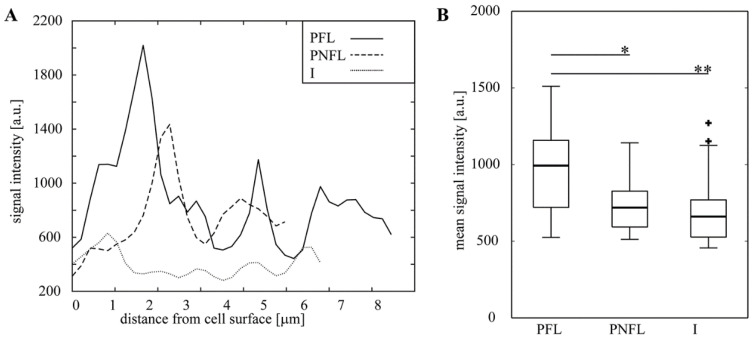
Calcofluor signal intensity in cell walls of lemma fibres. **A.** Signal intensity measured along transects marked in Figure 6. **B.** Mean signal intensities for cell walls of differently treated lemmas. Thick lines within boxes represent medians; boxes delimit the first and third quantiles; whiskers extend from each end of the box to the adjacent values in the data as long as the most extreme values are within 1.5 times the interquartile range from the ends of the box; crosses represent outliers. In each sample, walls of 18 cells from different portions of 2 lemmas were analyzed. Statistical significance is presented as: * *p* < 0.05; ** *p* < 0.01.

**Figure 8 biomolecules-11-00239-f008:**
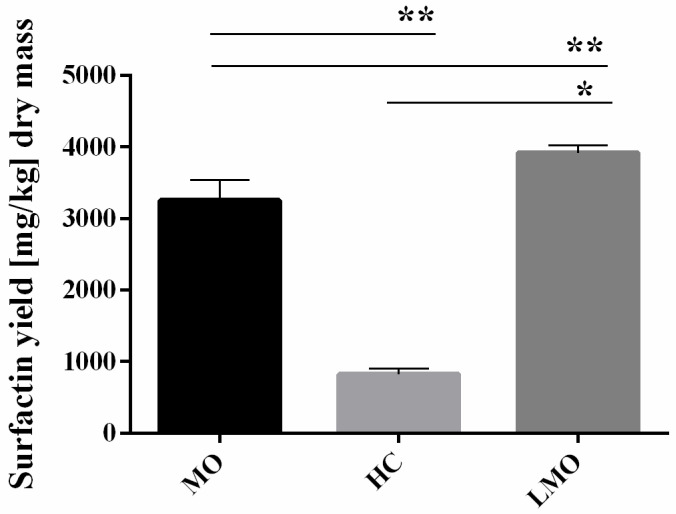
Surfactin yield obtained with *B. subtilis* 87Y during 24h of SSF on milled oats (MO), hulled caryopses (HC) and large fragments separated from milled oats (LMO). Statistical significance in all cases is presented as follows: * *p* < 0.05; ** *p* < 0.01.

**Table 1 biomolecules-11-00239-t001:** Gene specific primer sequences and amplification efficiencies.

Primer	Sequence [5’–3’]	Amplification Efficiency [%]
gyrB_F	TAATGGCGGCAAGAGCAAGA	102.08
gyrB_R	ATGTCTGTCGCGTCCTTGTT	
xynA_F	GCGAACCTGTAGTCCAACCTT	102.6
xynA_R	TTTTCGGCAACCGCCTCT	
xynB_F	AAACTGACAGAAGCTCCGCA	89.3
xynB_R	GGATTTCCTGGGTCATGCCA	
abfA_F	AGAGCCTTTCGGATGGTTGC	100.2
abfA_R	GACGGCTTGACTTGGCATG	
cah_F	TGCAGGCGATGAAGACACTT	86.16
cah_R	GCGGTACACCTTCAGCTCTT	

**Table 2 biomolecules-11-00239-t002:** Hemicellulose content of SSF medium used in the study. RSM—rapeseed meal, MO (milled oats) followed by a number specifies the level of RSM replacement by MO, NF—non-fermented, F—fermented. Hemicellulose loss is calculated as (NF–F)/NF in percent.

Solid Base	Hemicellulose [%]		Hemicellulose Loss [%]
	NF	F	
RSM_100	12.17	12.11	−0.49
MO_50	19.58	15.43	−21.19
MO_100	39.86	26.83	−32.68

## Data Availability

The data presented in this study are available on request from the corresponding author.

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
