# Peer review of "Xylan Decomposition in Plant Cell Walls as an Inducer of Surfactin Synthesis by Bacillus subtilis"

_biomolecules, 2021, doi:10.3390/biom11020239_

Round 1

Reviewer 1 Report

  1. Iturin and fengycin are the main lipopeptides having strong antifungal activities, while surfactin has antibacterial activity. In this manuscript, the authors should explain why they only discuss surfactin not also discuss iturin and fengycin? Is it the possible that xylan decomposition in plant cell walls as an inducer of iturin and fengycin synthesis by Bacillus subtilis?
  2. In the section of Materials and Methods, solid-state fermentation was performed only for 24h, the author should explain why not shown the time-course study. The yield of 48h may be better than 24h.
  3. line 15 and line 41 ”Bacillus subtilis 87Y strain” suggested correct to ”Bacillus subtilis strain 87Y” or ”Bacillus subtilis 87Y ”

Author Response

  1. Iturin and fengycin are the main lipopeptides having strong antifungal activities, while surfactin has antibacterial activity. In this manuscript, the authors should explain why they only discuss surfactin not also discuss iturin and fengycin? Is it the possible that xylan decomposition in plant cell walls as an inducer of iturin and fengycin synthesis by Bacillus subtilis?

Thank you for this suggestion. In our experimental conditions, we did not observe other lipopeptides than surfactin. We added this information in manuscript.

  1. In the section of Materials and Methods, solid-state fermentation was performed only for 24h, the author should explain why not shown the time-course study. The yield of 48h may be better than 24h.

Thank you. Indeed, surfactin yield of 48h potentially could be higher than after 24h. However, we aimed to investigate B. subtilis 87Y in terms of its potential industrial application, for which the 48h cultivation would be too long and economically unjustified.

  1. line 15 and line 41 ”Bacillus subtilis 87Y strain” suggested correct to ”Bacillus subtilis strain 87Y” or ”Bacillus subtilis 87Y ”

Thank you. We corrected the term, now it is “Bacillus subtilis 87Y” .

Reviewer 2 Report

The manuscript presented the study of hemicellulose components utilization by Bacillus subtilis. The results of product analysis, enzymatic activity and analysis of cell wall composition showed that xylan from hemicellulose is the main substrate for B. subtilis to produce surfactin.

The manuscript can be accepted for publication after minor revision:

-Introduction should be rewritten, specially from line 35-45 in which you explained about the aim of your study. Also, the explanation in line 46-52 should be moved to the conclusion part.

-Materials and methods

-Milliliters should be (mL) not (ml). 

-Line 73: should be air-dried sample.

-Line 76: The explanation regarding crucibles in unclear.

Results and discussion 

-Line329-331: the explanation was repeated, no need to write the results again.

Line 343: It would be good to compare the results of your study with other studies regarding the yield obtained by similar or different substrates such as glucose for example.

Author Response

-Introduction should be rewritten, specially from line 35-45 in which you explained about the aim of your study. Also, the explanation in line 46-52 should be moved to the conclusion part.

Thank you for pointing to this problem. We have rewritten large part of the Introduction, especially the lines that the Reviewer referred to.

-Materials and methods

-Milliliters should be (mL) not (ml). 

Thank you. We replaced “ml” by “mL”.

-Line 73: should be air-dried sample.

Thank you, we corrected this term here and in other places.

-Line 76: The explanation regarding crucibles in unclear.

Thank you. We apologize for misused term. We replaced this by ‘precipitates’.

Results and discussion 

-Line329-331: the explanation was repeated, no need to write the results again.

Thank you. We deleted this fragment.

Line 343: It would be good to compare the results of your study with other studies regarding the yield obtained by similar or different substrates such as glucose for example.

Thank you for this suggestion. We compared our results with those of other authors, in the last paragraph of section “Components of oat hulls induce surfactin production”.
